# Observed Trends in Thermal Stress at European Cities with Different Background Climates

**Dimitra Founda [1,*], Fragiskos Pierros [1], George Katavoutas [1] and Iphigenia Keramitsoglou [2]**

[1] Institute for Environmental Research & Sustainable Development (IERSD), National Observatory of Athens, GR-15236 Athens, Greece
[2] Institute for Astronomy, Astrophysics, Space Applications and Remote Sensing (IAASARS), National Observatory of Athens, GR-15236 Athens, Greece
[*] Correspondence: founda@noa.gr; Tel.: +30-2108-1091-32

**Abstract:** Intensification of extreme temperatures combined with other socioeconomic factors may exacerbate human thermal risk. The disastrous impacts of extreme weather during the last two decades demonstrated the increased vulnerability of populations even in developed countries from Europe, which are expected to efficiently manage adverse weather. The study aims to assess trends in the exposure of European populations to extreme weather using updated historical climatic data in large European cities of different local climates and a set of climatic and bioclimatic indices. Colder cities experience higher warming rates in winter (exceeding 1 °C/decade since the mid-1970s) and warmer cities in summer. Hot extremes have almost tripled in most cities during the last two or three decades with simultaneous advancing of hot weather, while northernmost cities have experienced an unprecedented increase in the heat waves frequency only during the last decade. Bioclimatic indices suggested a robust tendency towards less cold-related stress (mainly in cold cities) and more heat-related stress in all cities. A doubling or tripling in the frequency of heat-related 'great discomfort' was found in southern cities, while in the cities of northern Europe, heat-related 'discomfort' conditions are becoming increasingly more frequent and have nearly quadrupled during the last decade.

**Keywords:** Europe; hot extremes; heat waves; cold extremes; cold waves; bioclimatic indices; heat stress; urban thermal risk

## 1. Introduction

Climate change related to anthropogenic forcing has received outstanding scientific attention over the last few decades. Today, there is a remarkable consensus between scientists, governments, and the public that global warming constitutes a major threat for our planet, with devastating impacts on the environment, species, and societies, already experienced in most parts of the world [1]. Warming rates differ between different areas, seasons, or daily maximum ($T_{max}$) and daily minimum ($T_{min}$) air temperature [2]. According to the last IPCC Report (https://www.ipcc.ch/sr15/) it is of vital importance to limit global warming below 1.5 °C instead of the limit of 2 °C adopted in the Paris Agreement (https://unfccc.int/resource/bigpicture/#content-the-paris-agreemen) with respect to the pre-industrial levels, since the extra warming could pose much additional risk to ecosystems—possibly causing irreversible changes—and also to humans [3,4].

Changes in the mean climate are accompanied by changes in climatic extremes, which often evolve differently than mean values with even larger devastating effects [5–7]. The frequency of rare and dangerous weather phenomena like heat waves, floods, or unusually cold weather accelerates worldwide, constituting the most obvious manifestation of global warming effect. Heat waves (HWs

hereafter), in particular, are among the most widely experienced natural hazards during the last few decades, causing thousands of excess deaths in many parts of the world [8–11]. Anthropogenic climate change is reported to have more than doubled the odds for such events [12,13], while future projections indicate a further increase in their occurrence and severity [14–18].

Devastating impacts of extreme hot or cold temperature events on human health differ with respect to their specific characteristics such as their intensity and duration but also seasonal timing [19–21]. Early HWs or cold waves (CWs hereafter), for instance, are proven to have a stronger impact mainly due to lack of acclimatization of vulnerable population or lack of preparedness for extreme temperatures [22–24]. According to recent studies based on historical observations and/or future simulations, an earlier occurrence of extremely hot weather in the year is reported worldwide [17,25,26].

In addition to air temperature, other environmental parameters determine human sensation of thermal comfort or discomfort and, consequently, exposure to thermal risk, such as humidity, wind speed, or radiation. Observations and/or future simulations indicate that global increase in air temperature is accompanied with increase in specific humidity but also changes in maximum wet bulb temperatures [27–29]. Yet, the apparent temperature (AP) (the human-perceived equivalent temperature) has increased faster than air temperature over land especially in low latitudes and is expected to continue in the future, with the summertime increase in AP-based thermal discomfort outpacing the wintertime decrease in thermal discomfort [30]. It is expected that the simultaneous occurrence of higher air temperature and humidity could make climate conditions in some areas intolerable to humans in the future [28,31].

Exposure of populations to extreme temperatures and subsequent risk involves multiple contributing variables related to climatic and non-climatic factors. Socioeconomic components such as age, health status, poverty, adaptive capacity, access to cooling/heating systems, and others seem to largely influence the vulnerability of a population to thermal stress [17,32,33]. Demographic evolution, such as intensifying urbanization and population ageing, is expected to further exacerbate vulnerability to thermal stress in the near future [9]. By 2050, cities will be home for almost two-thirds of the global population [34]. It is known that urban environments further increase vulnerability to heat-related risk due to the urban heat island (UHI) effect and the synergistic interaction between HWs and UHIs [35,36]. Such synergies may increase the heat-related mortality rate in cities by four times when compared to rural areas [25,37]. Population ageing is also a global phenomenon. In 2017, 13% of the global population was aged above 60 and the percentage accelerates at a rate of 3% per year, with Europe sharing the greatest percentage of population aged above 60 (25%) [38].

Thermal stress and extreme weather have proven to be major threats even for developed countries, despite their higher standards of living and infrastructures [39]. From the beginning of the 21st century, Europe has witnessed a series of destructive extreme events with profound catastrophic impacts and very high death tolls, like the HWs of 2003 in Central Europe and 2010 in Moscow but also severe CWs in Northern and Western Europe in 2009/2010 [8,13,18,40]. In 2018, central and northern Europe experienced another prolonged period of drought and unusually hot weather. The northernmost municipality of Finland (Utsjoki), north of the Arctic Circle, experienced a record-breaking temperature of 33.3 °C in July 2018. The HWs caused increased elderly mortality, unprecedented wildfires in Scandinavian countries, and severe yields decline. It becomes clear that climate change affects all and the impacts are severe even in countries considered to be prepared for extreme weather management. Increasing population ageing and urbanization levels in Europe (reaching as high as 75% in 2015), along with regional warming, poses additional stress in European populations [9,40].

To better manage the adverse effects of extreme temperatures and consequent thermal stress in European populations, it is important to assess and quantify contemporary warming rates along with changes in extreme temperatures and other indices accounting for human body thermal sensation.

The present study focuses on seven European cities representative of different geographical areas and different background climatic characteristics, including southern, central, and northern Europe. The study was conducted in the framework of the EXTREMA (Extreme temperature alerts for Europe)

project, which aims to improve the resilience of the European population to extreme temperature events (https://extrema.space/). Based on updated historical observations of atmospheric variables spanning several decades and a set of climatic (air temperature-based) and bioclimatic indices accounting for human thermal comfort/discomfort standards, the study quantifies observed trends in mean climate, frequency of climatic extremes (hot or cold), and the level of exposure to heat- and cold-related stress (thermal stress) in the cities of interest. Although the study focuses on urban areas, it is beyond its scope to assess the causes behind the observed trends related to global climate change and/or increased UHI effect.

## 2. Data and Methods

### 2.1. Study Area

A number of European cities were selected for the study, as presented in the map of Figure 1. The cities represent different geographical areas of Europe, for instance southern, central, and northern Europe, and experience different background climatic conditions. Table 1 includes the cities along with coordinates and altitudes of the meteorological stations used for each city, as well as available study periods spanning several decades until the present. The climate type of each city, following the updated Köppen–Geiger climate classification [41,42], is also shown in Table 1. Largely varying climates between the selected cities are of interest, enabling the assessment of both heat- and cold-related risk and their long-term trends. It is also important to explore if and to what degree cold cities like Helsinki and Oulou encounter thermal discomfort conditions not previously experienced, or how heat-related stress evolves in warm cities like Athens. Large cities like Paris are also of interest due to the strong impact on population [8].

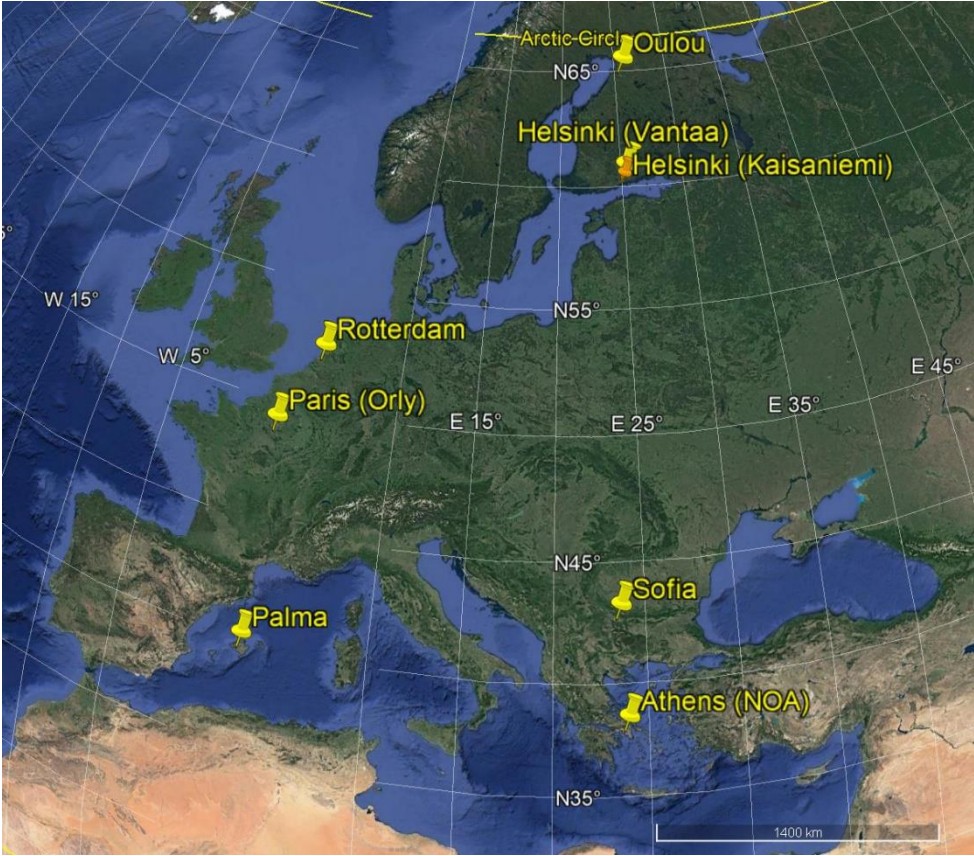

**Figure 1.** Map with the locations of the selected stations. (Map was produced in Google Earth Pro; Map data SIO, NOAA, US Navy, NGA, GEBCO; Image Landsat).

**Table 1.** List of selected stations along with coordinates, altitude, and study periods. The climate type for each city, following the Köppen–Geiger climate classification, is also presented.

| City/Station Name | Lat (°) | Long (°) | Alt (m) | Period | Climate Type (*) |
|---|---|---|---|---|---|
| Athens (NOA) | 37.972 | 23.717 | 107.0 | 1897–2018 | Csa |
| Palma | 39.550 | 2.733 | 8.0 | 1974–2018 | Csa |
| Sofia | 42.695 | 23.406 | 531 | 1972–2018 | Dfb |
| Paris (Orly) | 48.725 | 2.359 | 88.7 | 1948–2018 | Cfb |
| Rotterdam | 51.957 | 4.437 | −4.6 | 1956–2018 | Cfb |
| Helsinki (Kaisaniemi) | 60.175 | 24.948 | 4.0 | 1959–2018 | Dfb |
| Helsinki (Vantaa) | 60.317 | 24.963 | 54.6 | 1959–2018 | Dfb |
| Oulou | 64.930 | 25.355 | 14.3 | 1959–2018 | Dfc |

(*) C: Warm temperature, D: Snow, s: Summer dry, f: Fully humid, a: Hot summer, b: Warm summer, c: Cool summer.

In addition to the different background climates, the selected cities also differ with respect to other characteristics such as population (or rates of population growth), expansion of urbanized area, urban morphology, and others. Population ranges from nearly 200,000 (Oulou) or 400,000 (Palma) to approximately 3.5 million (Athens) or 11 million (Paris) including adjacent suburban areas (https://population.un.org/wup/). Yet, population growth at the selected cities is different over the study periods. The populations of Palma, Helsinki, and Oulou have more than doubled over the last four decades, while in other cities like Rotterdam, population has not increased significantly over the same period. Other differences between cities concern, for instance, different degree of sealed surfaces. Percentages of soil sealing per built up area are comparable between Athens, Sofia, and Paris, but values are different when comparing per inhabitant (https://www.eea.europa.eu/articles/urban-soil-sealing-in-europe). Helsinki has a lower percentage of soil sealing per built area but much higher sealed area per inhabitant compared to a city of similar population like Sofia, which exhibits the opposite pattern.

It becomes clear that the selected cities represent a variety of different climates and urban characteristics, thus, it is relevant and interesting to study climate variability and trends as well as related thermal risk at such different environments.

*2.2. Selection of Indices*

2.2.1. Climatic Indices

To assess the variability and trends in the mean climate on an annual and seasonal basis at the cities of interest, seasonal and annual means for each year were calculated from the mean daily, daily maximum ($T_{max}$), and daily minimum ($T_{min}$) air temperatures. A linear regression analysis was applied to estimate long-term trends in the data series, while the statistical significance of trends has been examined through the Mann–Kendall and Student's *t*-test.

For the definition of temperature extremes, a common procedure is the use of upper and lower percentiles of the probability distribution function; nevertheless, the impact of these extremes also depends on the baseline climate of the studied area [43].

On the other hand, defining HWs (or CWs) is still on open scientific issue. So far, different approaches have been adopted by researchers, varying with respect to the duration of the event, the selected climatic indices, or the absolute threshold values of temperature to determine extremes [44,45]. Yet, weather services worldwide adopt different definitions to issue alarms for heat-related risk. The large differences in the local climates at the cities of interest imposes the use of percentiles rather than fixed temperature thresholds for the definition of extremes. In this respect, the following indices were adopted for the study of extremes:

- Hot extreme (Hot Day): $T_{max} >$ 95th percentile of the summer daily $T_{max}$ distribution;
- Cold extreme (Cold Day): $T_{min} <$ 5th percentile of the winter daily $T_{min}$ distribution;

- Heat Wave (HW): Sequence of at least 3 consecutive Hot Days;
- Cold Wave (CW): Sequence of at least 3 consecutive Cold Days.

The choice of 3 days for the duration of HWs/CWs waves is in line with (or stricter than) other limits set worldwide [9,44,46].

### 2.2.2. Bioclimatic Indices

A number of additional environmental factors such as ventilation or relative humidity combined with other personalized factors (e.g., age, clothing, activity, time of exposure, and others) strongly influence human thermal comfort. Physiological acclimatization but also psychological adaptation or expectations may also affect thermal perception. There exists a plethora of bioclimatic indices in the literature aiming at the assessment of thermal environment and the impact on humans. The following indices were selected for the study, suitable for hot and/or cold stress.

- Heat index—HI (valid for heat stress);
- Humidex—HD (valid for heat stress);
- Effective temperature—ET (valid for heat and cold stress);
- New wind chill equivalent temperature—WCT (valid for cold stress).

The selected indices are broadly used in the literature but are also used by weather services for thermal stress-related alarms [47,48]. Selection criteria were also based on the availability of meteorological data necessary for the calculation of the indices.

Brief descriptions of the indices along with scales of degree of thermal comfort/discomfort are provided in the Appendix A.

The selected bioclimatic indices were calculated for each city based on available synchronous measurements of the involved meteorological variables at a resolution of 3 h. Time series spanning at least four decades were created for each index and city. The scales and threshold values corresponding to different comfort/discomfort levels are applied in all cities to portray thermal sensation, despite the different local climates. However, t is known that thermal perception is largely influenced by physiological acclimatization and adaptation, and local climate plays a major role in the perception of hot and cold conditions. For instance, an ambient temperature of 26 °C has been classified as 'hot' in Finland [49]; nevertheless, such temperatures may be considered normal or even cool in other regions [50].

### 2.3. Data

Meteorological observations for Athens were derived from the historical climatic record of the National Observatory of Athens (NOA). The record dates back to the end 19th century, enabling the study of climatic variability and trends in climatic extremes on centennial scale. The station is located on a small hill at the center of the city, isolated from densely built areas. Daily values of meteorological variables for Finland were extracted from the Finnish Meteorological Institute (http://en.ilmatieteenlaitos.fi/weather-and-sea) for the stations Helsinki-Vantaa (hereafter Helsinki-V) and Helsinki-Kaisaniemi (hereafter Helsinki-K). Helsinki-K represents an urban station at a very short distance from the sea, while Helsinki-V is the airport station to the north of the city, also affected by urban environment. Daily air temperature data for other cities were acquired from the database of KNMI Institute (https://eca.knmi.nl) [51] and refer to airport stations. Data include daily averages ($T_{avg}$), $T_{max}$, and $T_{min}$ air temperature.

Except for air temperature, additional meteorological parameters such as wind speed, relative humidity, and water vapor were utilized for the calculation of the bioclimatic indices. Synchronous meteorological observations for the selected sites at 3-hour resolutions were extracted from the database of the National Oceanic and Atmospheric Administration (NOAA) (https://www7.ncdc.noaa.gov/CDO/cdopoemain.cmd?datasetabbv=DS3505&countryabbv=&georegionabbv=&resolution=40). Only

Helsinki-V and Oulou were used in the analysis of bioclimatic indices in Finland, following the availability of observations at 3-h resolution.

The periods of available data differ between the selected stations. Depending on the kind of analysis and the examined climatic (or bioclimatic index), different periods are studied based also on data completeness and reliability for the particular analysis. In all cases, periods span at least four decades, namely from the mid-1970s until the present. Actually, this period reflects the period of the most prominent warming in the Northern Hemisphere, while 1976 is a widely acknowledged 'climate shift' year [52,53].

## 3. Results

### 3.1. Mean Climatic Values of Air Temperature at the Selected Cities

Table 2 shows the seasonal and annual mean climatic values of the daily average temperature ($T_{avg}$) at the selected cities over the commonly available period 1974–2003. These values demonstrate the markedly different background climates at the cities of interest. Athens and Palma are the warmest cities during all seasons, while Oulou and Helsinki-V the coldest ones. Seasonal variability is more pronounced in northern cities (Helsinki, Oulou) and Sofia, where differences between winter and summer mean temperature are at least 20 °C. The different background climates at the selected cities are further highlighted in the values of the upper (95th) percentile of the distribution of the summer daily maximum temperature ($T_{max}$) and the lower (5th) percentile of the distribution of the winter daily minimum temperature ($T_{min}$) for each city, also shown in Table 2. The percentiles were also derived from the common period 1974–2003. The differences in the percentiles between the stations may exceed 11 °C in summer (e.g., between Athens and Helsinki) but are striking in winter, reaching up to 30 °C (e.g., between Athens and Oulou).

**Table 2.** Seasonal and annual mean climatic values of the daily average temperature ($T_{avg}$) at the selected cities for the common period 1974–2003, along with standard deviations. Upper and lower percentiles of the summer/winter $T_{max}/T_{min}$ temperature distributions over the same period for each city are also presented.

| City/Station Name | Winter (°C) | Spring (°C) | Summer (°C) | Autumn (°C) | Year (°C) | 95th Percentile (°C) | 5th Percentile (°C) |
|---|---|---|---|---|---|---|---|
| Athens (NOA) | 9.92 (±0.91) | 15.83 (± 0.98) | 26.61 (± 1.02) | 18.82 (± 0.82) | 17.79 (± 0.57) | 37.1 | 2.0 |
| Palma | 9.95 (±0.99) | 13.70 (±0.90) | 23.48 (±1.06) | 17.83 (±0.85) | 16.25 (±0.67) | 35.0 | −1.4 |
| Sofia | −0.09 (±1.22) | 9.90 (±0.1.16) | 19.50 (±0.99) | 10.43 (±1.00) | 9.93 (±0.52) | 33.0 | - |
| Paris (Orly) | 4.47 (±1.13) | 10.77 (±0.97) | 18.87 (±1.24) | 11.89 (±0.75) | 11.49 (±0.76) | 31.7 | −5.1 |
| Rotterdam | 3.80 (±1.76) | 9.09 (±1.01) | 16.69 (±0.99) | 10.83 (±0.83) | 10.10 (±0.78) | 28.6 | −7.6 |
| Helsinki (Kaisaniemi) | −2.74 (±0.99) | 3.99 (±1.13) | 15.96 (±1.05) | 6.28 (±1.16) | 5.58 (±1.03) | 25.5 | −19.8 |
| Helsinki (Vantaa) | −5.03 (±2.87) | 3.86 (±1.18) | 15.67 (±1.16) | 5.28 (±1.21) | 4.94 (±1.12) | 27.3 | −22.4 |
| Oulou | −9.1 (±3.16) | 1.33 (±1.26) | 14.51 (±1.00) | 2.78 (±1.38) | 2.38 (±1.08) | 26.2 | −28.5 |

### 3.2. Mean Air Temperature Trends

Figure 2 depicts the long-term variation of the annual mean air temperature (anomalies of the annual values from the 1974–2003 mean) at representative cities over the available study period for

each city. All curves exhibit a fluctuating pattern with alterations between warmer and colder periods over the years. This is particularly evident in the centennial time series of Athens, where for instance, warmer conditions in the 1950s are followed by a cooler period until the mid-1970s, when an ongoing warming is observed until the present. Despite the fluctuating patterns, a statistically significant ($p < 0.05$) long-term upward trend was observed in all cities denoting the warming tendency over the study periods. All stations experience an ongoing warming from the mid-1970s onwards, amounting roughly to 0.4 °C/decade at Paris and Rotterdam; 0.5 °C/decade in Athens, Palma, and Sofia; and 0.6 °C/decade at Helsinki.

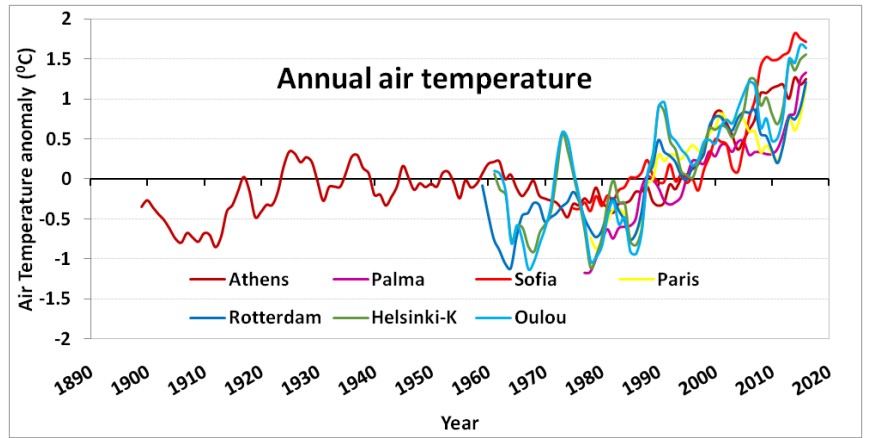

**Figure 2.** Long-term variation of the annual mean air temperature anomaly (difference from the 1974–2003 mean) at selected European cities. A five-year moving average filter was applied.

Seasonal analysis of the daily maximum ($T_{max}$) and daily minimum ($T_{min}$) temperatures disclosed markedly different warming rates depending on the season, index, and city. Figure 3a,b shows the trends (slopes and confidence intervals derived from linear regression analysis, in °C/year) of $T_{max}$ and $T_{min}$ at all sites over the common period 1976–2018. Due to large gaps in daily records of $T_{min}$ at the Sofia station, the seasonal trends were derived from monthly values of $T_{min}$ calculated from the broadly used formula $T_{avg} = (T_{max} + T_{min})/2$ [54]. All trends are positive (except for the winter trend of $T_{min}$ at Sofia, which is marginally negative) indicating the warming tendency in both daytime and nighttime air temperature in European cities over the last decades. Moreover, the vast majority of trends are statistically significant at least at 95% significance level (Tables A5 and A6 in Appendix B). However, the warming rates exhibit significant inter-diurnal (difference between $T_{max}$ and $T_{min}$), seasonal, and spatial variability between different European areas.

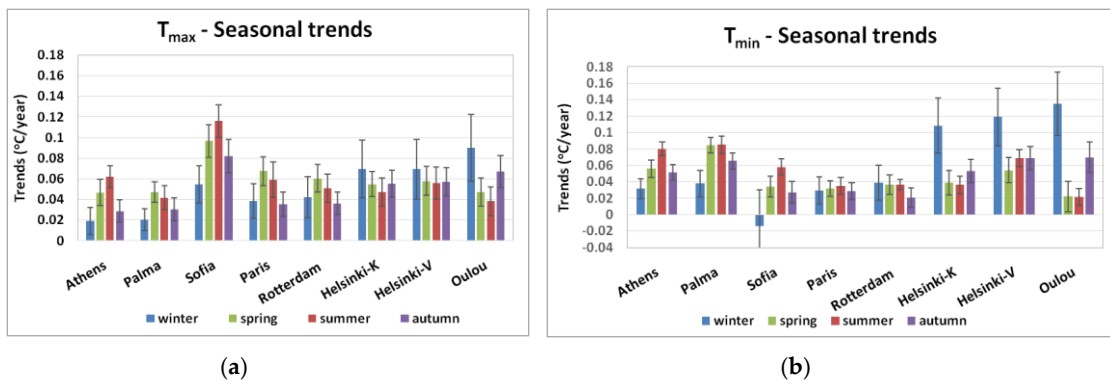

| (a) | (b) |

**Figure 3.** Seasonal linear trends of $T_{max}$ (**a**) and $T_{min}$ (**b**) at the selected European cities (1976–2018), along with standard errors. Statistical significance has been tested using the Mann–Kendall trend test and *p* values are presented in Tables A5 and A6.

Cities at higher latitudes like Helsinki and Oulou exhibit particularly large warming trends in both $T_{max}$ and $T_{min}$ during the cold period of the year. Specifically, the warming rates of $T_{min}$ in winter is outstanding, exceeding 1 °C per decade. Lower but statistically significant positive trends are also observed in spring and summer. Helsinki has been experiencing significant warming in summer $T_{max}$, amounting to about half a degree per decade (or approximately 2 °C since the mid-1970s at Helsinki-V). Notably, Helsinki-V presents higher warming rates than Helsinki-K, also reflecting the effect of local scale factors on temperature trends. In contrast, southeastern cities like Athens and Sofia present higher warming rates during the warm season of the year compared to the cold season. Sofia in particular experiences the highest warming rates in summer compared to any other city, reaching up to 1.2 °C per decade. This resulted in an outstanding total increase in summer $T_{max}$ by 4–5 °C since the mid-1970s in the city. A pronounced increase in summer $T_{min}$ is observed in Athens (~0.8 °C/decade), as a result of both regional warming and urban heat island effect [36,55,56].

The year 2018 was the warmest year of the historical record of $T_{min}$ in Athens; Paris, Palma, and Rotterdam experienced higher warming rates in spring and summer $T_{max}$, while seasonal variability in $T_{min}$ trends are less important, except for Palma, which presents very large warming trends in the summer and spring nighttime temperature, corresponding to an increase of the order of 3–4 °C over the whole study period.

### 3.3. Trends in the Frequency of Hot and Cold Extremes

In the following, we present the variability and trends in the frequency of hot and cold extremes as defined in Section 2.2.1, at the selected cities. Figure 4 depicts the frequencies (total number) of hot and cold days per decade (left panels) and frequencies (total number) of HWs and CWs per decade (right panels) over the available study period for each station.

The analysis of the hot extremes revealed that the observed warming trends in the mean climate (Figures 2 and 3) have been accompanied with concurrent positive trends in the frequency of hot extremes. All cities experienced increases in the frequency of hot extremes (hot days and HWs) during the last two or three decades. Slopes of the observed trends over the entire available period for each station are also presented in Table A7 (Appendix B).

The total number of hot days and HWs per decade in Athens and Sofia has increased nearly threefold or fourfold after the late-1990s. Other cities like Paris and Rotterdam have also experienced almost a tripling in the hot extremes' frequency since the late 1980s. At Helsinki-V, the frequency of hot days has been increasing progressively since the 1980s; nevertheless, a very prominent increase is observed in the last decade (2009–2018). A particular feature at Helsinki-V is the abrupt rise in the frequency of HWs just in the last decade in contrast to other cities where HWs have become more frequent one or two decades earlier. Similar results were also found at Helsinki-K and Oulou (not shown).

Warming trends were also associated with simultaneous decreasing trends in the occurrence of cold extremes, namely cold days and CWs (Figure 4, Table A7). Almost all cities have experienced a statistically significant negative trend in the frequency of cold days and/or CWs over the study periods. The number of cold days decreased almost progressively since the mid-20th century in Paris, Rotterdam, and Palma, while at Helsinki, the decrease is more abrupt after the late 1980s. Until the late 1980s, the frequency of cold extremes exceeds the frequency of hot extremes almost in all cases, while this pattern reverses afterwards, with the frequency of hot extremes exceeding cold ones. Frequency of CWs at northern cities was found to be almost half during the last three decades compared to the previous ones following a constant decreasing tendency, in accordance with the marked increasing trends in winter $T_{min}$ (Figure 3). Frequency of CWs in Athens shows a fluctuating pattern over the past century, with overall negative tendency. A remarkably higher frequency in such events is observed during the cold period spanning the late 19th to early 20th century. Noticeably, despite the upward trend in the annual temperature in Athens since the mid-1970s (Figure 2), the cold extremes do not show a clear decline over the same period, which is also consistent with the relatively lower trends in winter time (Figure 3).

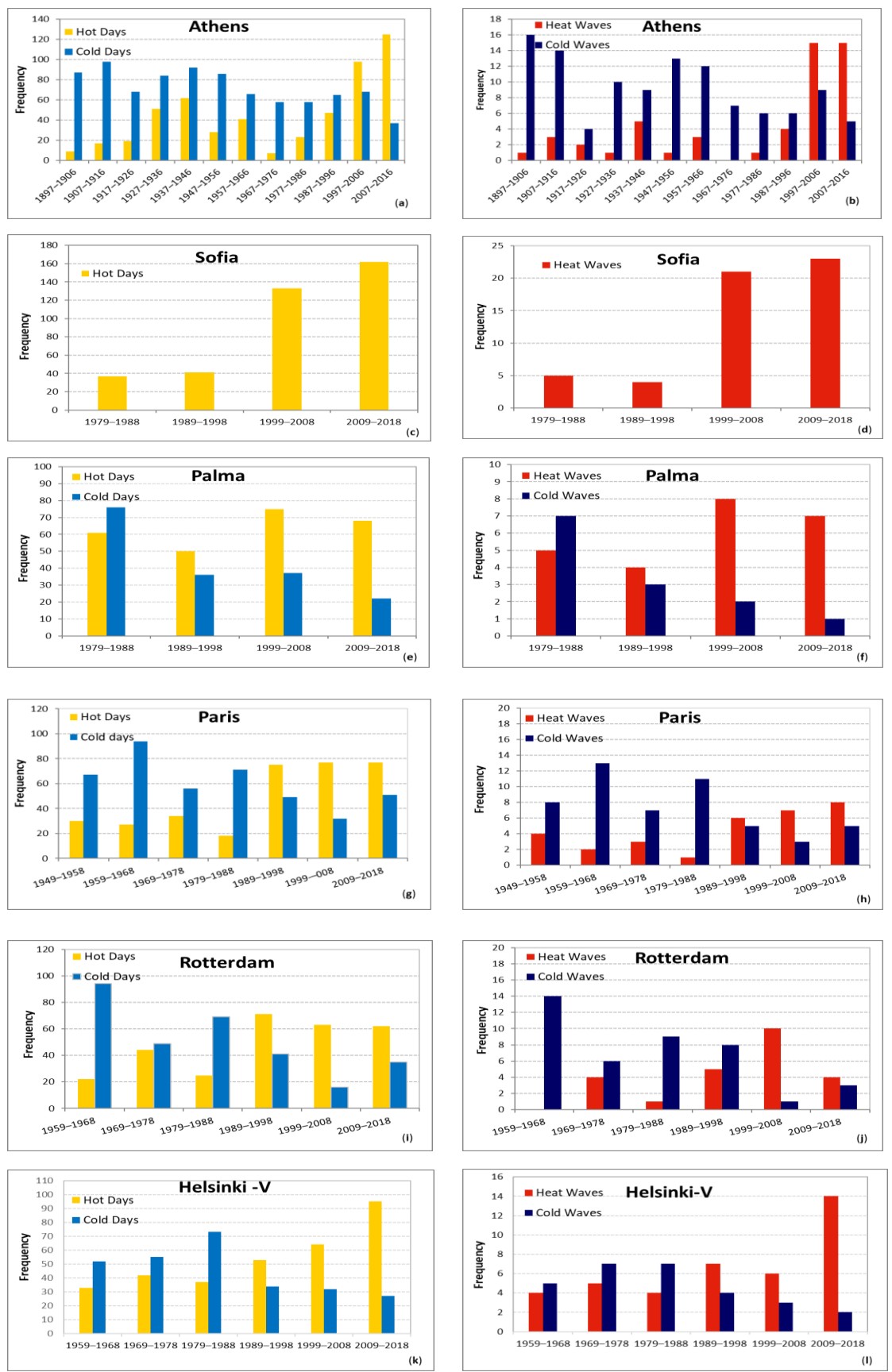

**Figure 4.** Frequency (total number) of hot/cold days per decade in (**a**) Athens, (**c**) Sofia, (**e**) Palma, (**g**) Paris, (**i**) Rotterdam, (**k**) Helsinki-V and heat waves (HWs)/cold waves (CWs) per decade in (**b**) Athens, (**d**) Sofia, (**f**) Palma, (**h**) Paris, (**j**) Rotterdam, (**l**) Helsinki-V, over the study period for each station.

### 3.4. Trends in the Timing and 'Season' of Hot Extremes

Not only the frequency or intensity of hot extremes, but also their seasonal 'timing', namely the dates of their occurrence in the year, is of high importance as it has proven to impact human health and influence rates of mortality/morbidity [22–24]. The Julian Day (number of days from 1 January) corresponding to the occurrence of the first and last hot extreme in the year was calculated for each year and station [25,26]. Table 3 presents the trends in the timing (Julian Day) of the first and last hot extreme at the selected cities over the common period 1976–2018. Negative trends in the date of the first hot extreme imply earlier onset and positive trends delayed the onset of hot extremes' season. Accordingly, negative/positive trends in the date of the last hot extreme indicate earlier/delayed ending of the hot extremes' season.

**Table 3.** Trends in the timing of the first and last hot extreme at selected cities (1976–2018). Bold values denote statistical significance at 95% level ($p < 0.05$).

| City | First Hot Extreme (Days/Year) | Last Hot Extreme (Days/Year) |
|---|---|---|
| Athens | **−0.436** | **+0.556** |
| Palma | **−0.583** | +0.056 |
| Sofia | **−0.823** | **+0.991** |
| Oulou | −0.238 | +0.108 |
| Helsinki-V | **−0.713** | +0.130 |
| Paris | **−0.833** | +0.133 |
| Rotterdam | −0.286 | +0.133 |

Results in Table 3 indicate an almost consistent, robust pattern in all cities, characterized by negative trends in the date of the first hot extreme and positive trends in the date of the last hot extreme. Dates of hot extremes have shifted earlier in the year by almost eight days/decade in Sofia, Paris, and Helsinki, and by four to six days/decade in Palma and Athens. In other words, today, hot extremes come sooner by half to one month compared to the mid-1970s. Accordingly, the last hot extremes are shifted later in the year at all cities. These later shifts are prominent in Sofia (nine days/decade) and Athens (five days/decade) while other stations experience smaller, not statistically significant trends.

Earlier temporal shifts of the first hot extreme along with later temporal shifts of the last hot extreme induce lengthening of the hot extremes' season, namely the time interval (number of days) between the dates of the first and last extreme for each year [25,26]. Simultaneous earlier/later shifts of the first/last hot extreme in southern stations resulted in a profound ongoing expansion of the hot extremes' season at both sides (beginning and ending of summer), which approximates 20 days/decade in Sofia and 13 days/decade in Athens (Figure 5a,b). Lower rates were found in Paris, Palma, and Rotterdam, ranging from four to eight days/decade, while in northern stations, the length of the hot extremes' season almost doubled during the last decade.

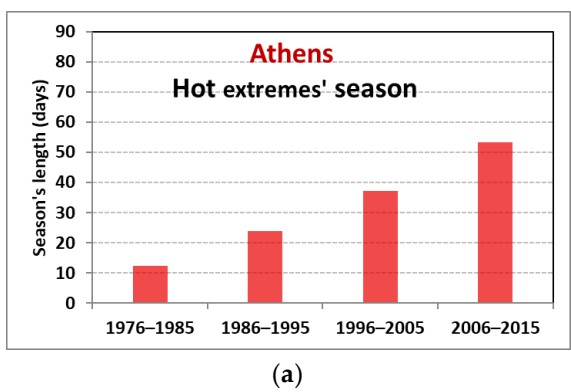

(a)

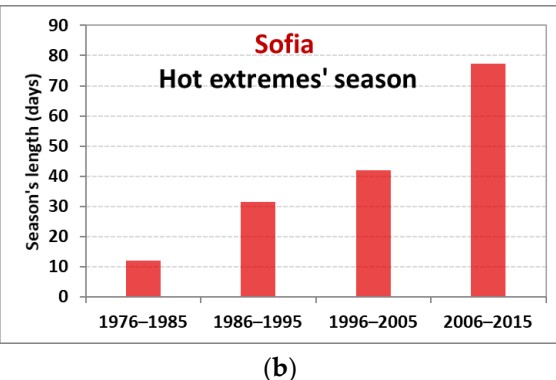

(b)

**Figure 5.** Average length of the hot extreme's season in Athens (**a**) and Sofia (**b**) for the past four decades.

### 3.5. Trends in Thermal Discomfort Based on Bioclimatic Indices

3.5.1. Trends in the Upper and Lower Percentiles of Bioclimatic Indices

Based on the levels of heat- or cold-related discomfort conditions suggested by the scales of the relevant indices (Tables A1–A4), it is interesting to examine possible trends in the upper (related to high heat-related risk) or lower (related to high cold-related risk) percentiles of the corresponding index. Positive (negative) trends in the upper percentile of the valid values for heat stress indicate worsening (improvement) in the heat-related thermal discomfort [56]. Accordingly, positive (negative trends) in the lower percentile of the valid values for cold stress indicate improvement (worsening) in the cold-related thermal discomfort.

Table 4 presents the trends in the upper (98th) and lower (2nd) percentiles of the distribution of all bioclimatic indices for each city over the period 1976–2018. Results indicate that both the 98th and 2nd percentiles of all indices and at any city reveal positive trends in the long-term. The positive trends in the upper percentile for the heat-related indices suggest a progressive worsening of thermal discomfort over the past decades. This is more prominent in the case of Athens and Sofia. Figure 6a,b also depicts the long-term trend in the 98th percentile of the HI, HD, and ET for Athens and Sofia. Regarding the cold-related stress, the positive trends in the lower 2nd percentile of the valid values for cold stress in ET and WCT imply an improvement in the cold-related stress over the past decades. This is more prominent at the cities with colder background climate like Rotterdam, Oulou, and Helsinki, also depicted in Figure 6c,d.

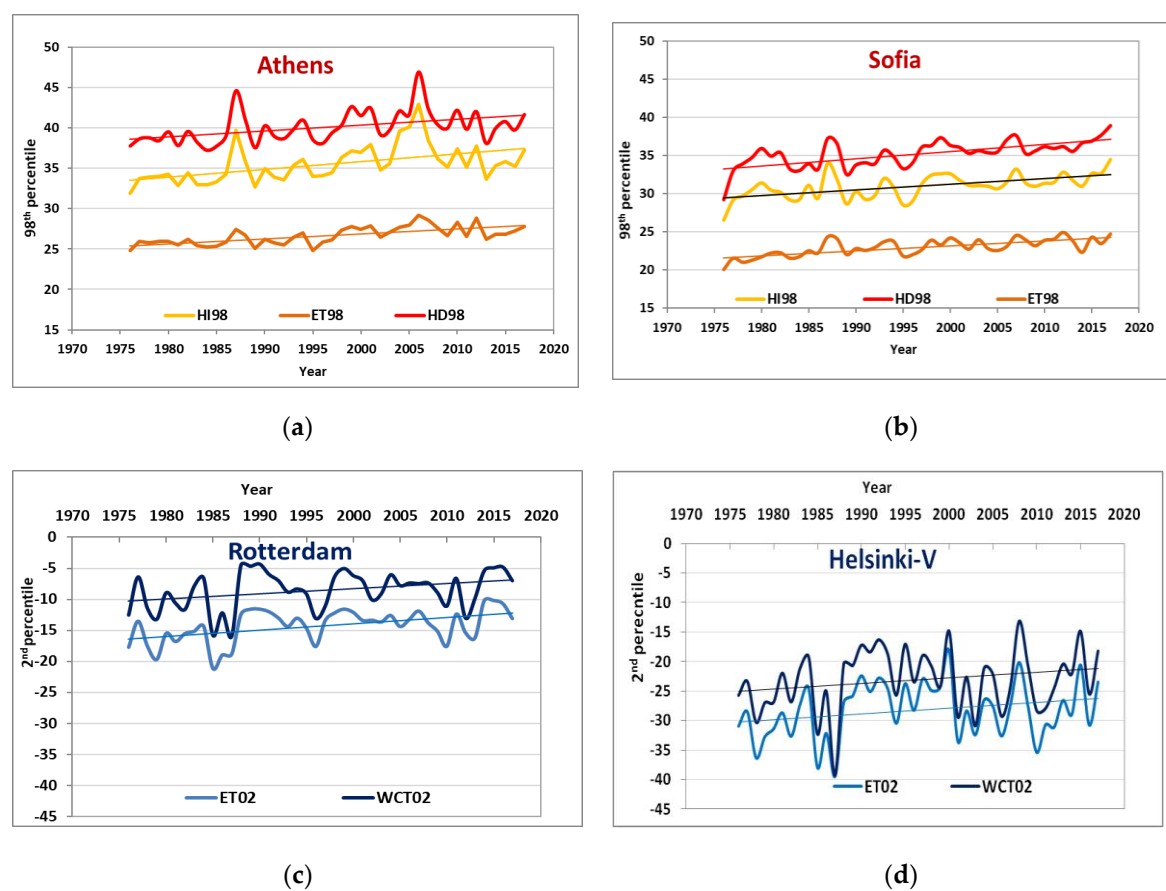

**Figure 6.** Temporal variation of the upper 98th of heat index (HI), humidex (HD), and effective temperature (ET) at Athens and Sofia (**a**,**b**) and of the lower 2nd percentile of ET and new wind chill equivalent temperature (WCT) at Helsinki-V and Rotterdam (**c**,**d**).

**Table 4.** Trends in the upper percentile of valid for heat stress indices and lower percentile of valid for cold indices at the selected cities (1976–2018). Bold characters indicate statistically significance at 95% confidence level.

| City | 98th Percentile–HOT | | | 2nd Percentile–COLD | |
|---|---|---|---|---|---|
| | HI | HD | ET | ET | WCT |
| Athens | **+0.097** | **+0.073** | +0.062 | +0.030 | +0.016 |
| Palma | +0.024 | +0.015 | +0.026 | +0.030 | +0.031 |
| Sofia | **+0.076** | **+0.094** | +0.065 | +0.045 | +0.066 |
| Paris | +0.043 | +0.042 | +0.043 | +0.063 | +0.042 |
| Rotterdam | +0.039 | +0.035 | +0.041 | **+0.105** | **+0.082** |
| Helsinki-V | +0.026 | +0.053 | +0.023 | **+0.092** | **+0.098** |
| Oulou | +0.013 | +0.027 | +0.040 | **+0.095** | **+0.072** |

Moreover, it was found that according to WCT scale of thermal sensation (Table A4), the frequency of cold-related discomfort conditions (cases corresponding to WCT < −10 °C) in northern cities (Helsinki-V and Oulou) dropped by more than 20% in the last decade compared to the decade 1976–1985. The decline is much higher in risk or high-risk levels (WCT < −28 °C) ranging from 36% at Oulou to 45% at Helsinki, also in agreement with the increasing trends in the 2nd percentile corresponding to high risk values (Figure 6). The frequency of discomfort conditions (WCT < −10 °C) lessened by nearly 60% in Rotterdam and by 47% in Paris during the last decade compared to the 1970s. Corresponding frequencies for cold-related discomfort conditions are negligible at Athens and Palma.

### 3.5.2. Trends in the Heat-Related Discomfort Conditions

The previous analysis based on the upper/lower percentiles of the indices suggested worsening towards increased heat-related discomfort at all cities, better highlighted from the use of the HD and HI indices, valid for heat stress. Hence, the following analysis focuses on a more detailed examination of the long-term changes and trends in different levels of thermal discomfort based on HD and HI. The frequency (number of cases at the three-hour resolution) when the index value exceeds certain thresholds was calculated for each year and city.

The inter-annual variability of the frequencies along with linear trends for each city and index are depicted in Figures 7 and 8. In this respect, based on HI classification, at least 'caution' corresponds to all cases when HI > 27 °C and at least 'extreme caution' to all cases when HI > 32 °C. Note that although HI and HD are expressed in °C, these values do not correspond to air temperature values. As for HD classification, at least 'discomfort', 'great discomfort', and 'danger' conditions correspond to all cases when HD values exceed 29, 39, and 45 °C, respectively.

Based on HI classification (Figure 7), all cities were found to experience statistically significant ($p < 0.05$) upward trends in the frequency of at least 'caution' and at least 'extreme caution' conditions over the years, except Rotterdam in at least 'caution' and Paris in at least 'extreme caution' categories, with $p < 0.10$. As expected, noticeably higher frequencies are observed in cities with hotter background climate in summer like Athens, Palma, and Sofia. In Athens and Palma, the frequency of at least 'caution' and 'extreme caution' conditions exhibit the largest trends, while both categories have almost tripled in Sofia during the last decade compared to the mid-1970s. In the cities of higher latitudes like Oulou and Helsinki, 'caution' conditions are becoming increasingly more frequent during the last decade. Such conditions are almost missing in the 1970s, while 'extreme caution' cases are not observed.

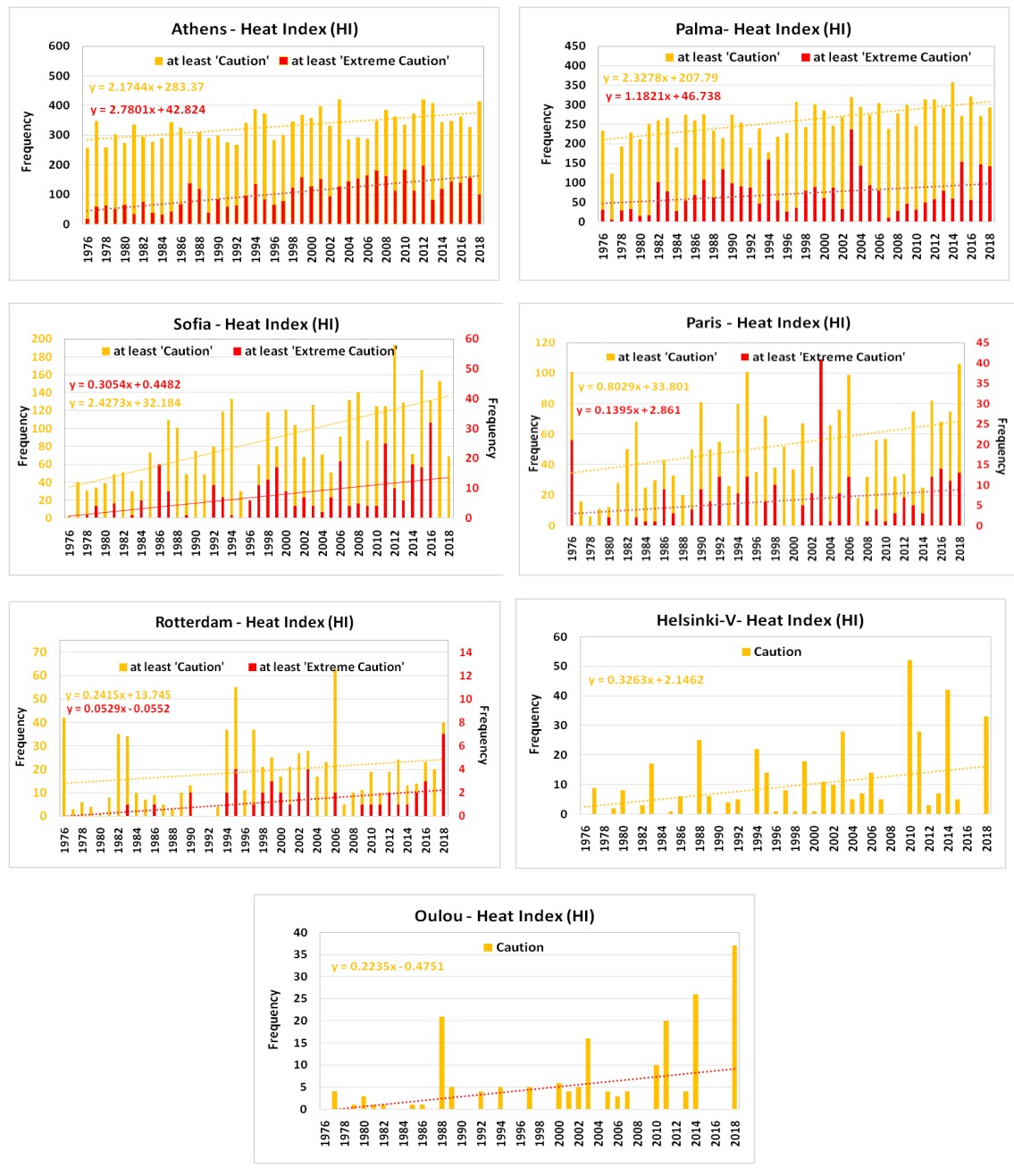

**Figure 7.** Frequencies (number of cases from the three-hour resolution observations) of at least 'caution' (HI > 27 °C) and at least 'extreme caution' (HI > 32 °C) conditions based on the heat index (HI) classification at European cities over the period 1976–2018.

The results based on HD classification (Figure 8) are in close qualitative agreement with HI, with frequencies of heat stress conditions exhibiting upward trends in all cities and levels of heat stress The trends in the frequency of at least 'discomfort' conditions (HD > 29 °C) are statistically significant in all cities ($p < 0.05$). Higher values of HD corresponding to the level 'danger' were observed occasionally only at the cities of Athens and Palma, mainly during the severe HWS in 2003 (in Palma) and 1986 and 2007 in Athens (not shown).

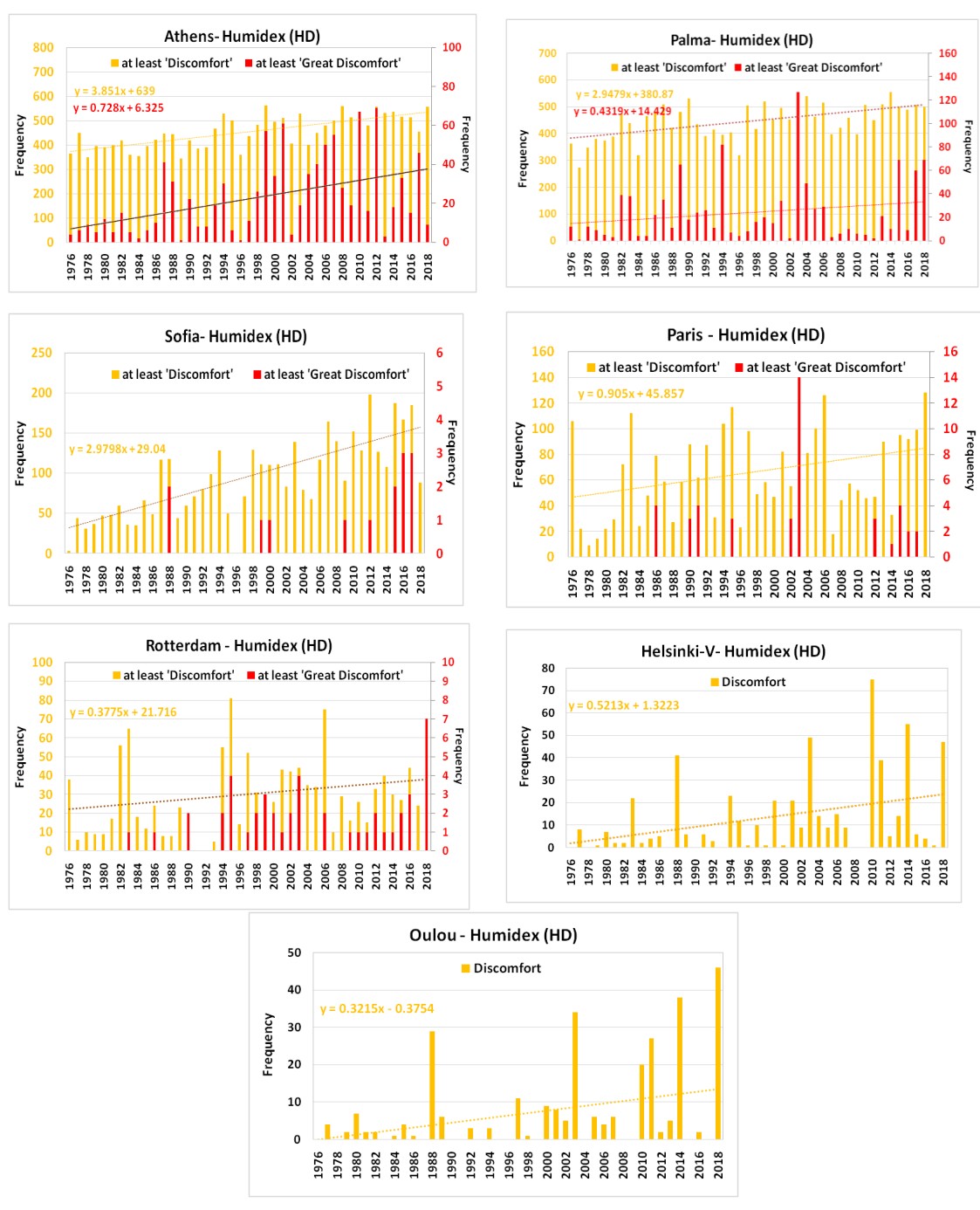

**Figure 8.** Frequencies (number of cases from the three-hour resolution observations) of at least 'discomfort' (HD > 29 °C) and at least 'great discomfort' (HD > 39 °C) conditions based on the humidex (HD) classification at European cities over the period 1976–2018.

At Oulou and Helsinki-V, 'discomfort' conditions are observed in certain years, but their frequency is noticeably higher during the last decade. Such conditions were very rare until the late 1980s, while 'great discomfort' cases are not observed.

The HW of 2003 in central Europe is clearly captured by both indices and is portrayed in the frequency of 'great discomfort' and 'extreme caution conditions' in Paris and Palma, while the extremely hot conditions that prevailed in northern and central Europe during summer 2018 are illustrated in the corresponding discomfort conditions in northern cities and particularly in Oulou, but also in Paris and Rotterdam.

### 3.5.3. Cumulative Distribution Functions (CDFs) of Bioclimatic Indices

Bioclimatic indices suggest levels of human comfort/discomfort based on certain threshold values (or ranges between threshold values) of the index. In this respect, it is useful to produce the cumulative distribution function (CDF) of the indices, as they are helpful to estimate probabilities above or below a certain value, or between two values. Some examples from the application of CDF curves of the bioclimatic indices are presented in Figure 9a–j. Observed shifts to the right and less steep CDF curves imply increased probabilities above certain threshold values related to increased heat stress conditions (Figure 9a–f).

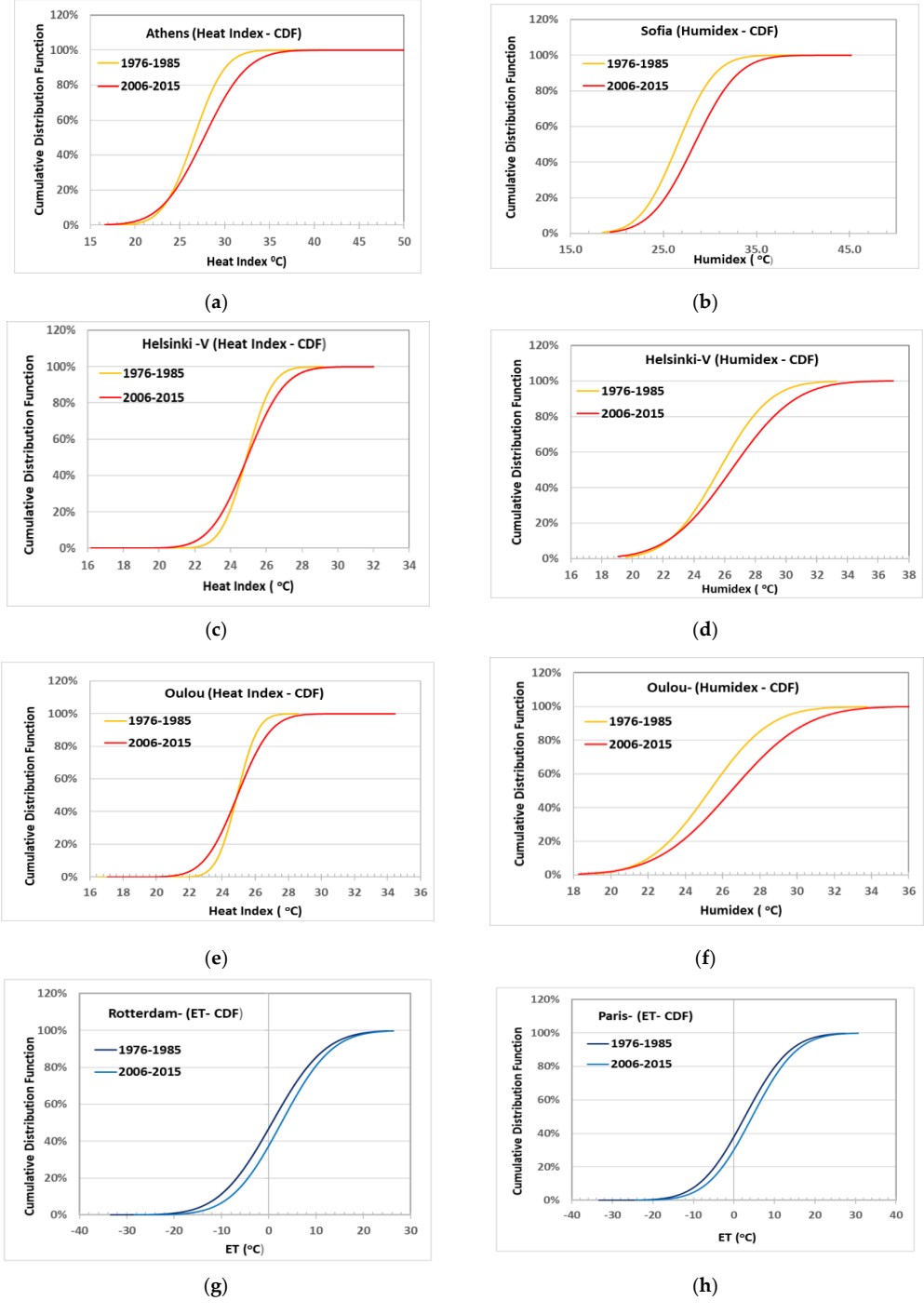

**Figure 9.** *Cont.*

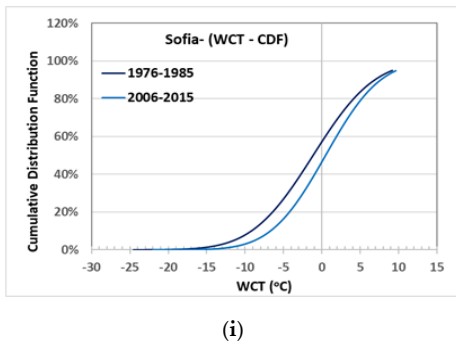 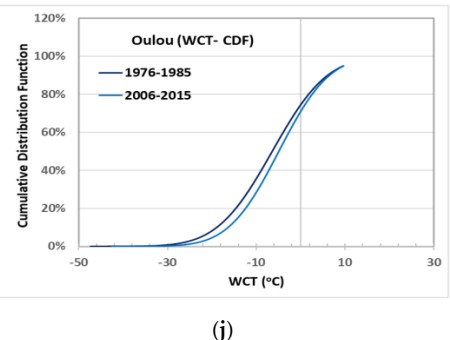

(**i**)                                             (**j**)

**Figure 9.** Curves of the cumulative distribution function (CDF) of the humidex (HD), heat index (HI), effective temperature (ET), and new wind chill equivalent temperature (WCT) at different European cities, calculated for the two decades 1976–1985 and 2006–2015. CDFs of (**a**): HI for Athens, (**b**): HD for Sofia, (**c**): HI for Helsinki-V, (**d**): HD for Helsinki-V, (**e**): HI for Oulou, (**f**): HD for Oulou, (**g**): ET for Rotterdam, (**h**): ET for Paris, (**i**): WCT for Sofia, (**j**): WCT for Oulou.

Figure 9a depicts CDF curves of the calculated HI in Athens for the two decades of 1976–1985 and 2006–2015. A shift of the CDF curve to the right is observed during the recent decade suggesting the increased probabilities of higher values throughout the range of the index. Following the HI scale, the probability of at least 'caution' conditions (HI > 27 °C) has increased from 43% in 1976–1985 to 58% in 2006–2015, while the corresponding 'extreme caution' probability (HI > 32 °C) has increased from 2.5% to 13%, respectively, indicating a fivefold increase in extreme caution conditions in Athens since the mid-1970s. Analogous increase was also found in the probability of at least 'great discomfort' conditions from the application of HD (not shown). Similarly, the probability of heat-related 'discomfort' conditions (HD > 29 °C) increased from 21.7% in 1976–1985 to 43% in 2006–2015 in Sofia (Figure 9b). Northern cities experience similar shifts in CDF curves, for instance the change in the probability of at least 'caution' conditions from 3.5% to 11% between the first and last decade at Helsinki-V (Figure 9c) and from 2% to 10% at Oulou (Figure 9e). Probability of at least 'discomfort' conditions based on HD was also found to have increased from 7.5% to 21% at Oulou and from 10% to 22% at Helsinki-V between the two periods (Figure 9d,f). Shifts in CDFs of HI and HD at Palma and Rotterdam were less prominent, while in Paris, the probability of at least 'caution' condition based on HI increased from 20% to 29% between the beginning and ending of the study period.

CDF curves were also produced for the indices related to cold stress, namely the effective temperature (ET) index and the new wind chill equivalent temperature (WCT) index (Figure 9g–j). Figure 9g shows the CDF curves of the ET index corresponding to the decades at the beginning and ending of the study period for Rotterdam. The CDF curve in the last decade also exhibits a shift to the right, namely to higher values of the index. Following the scales for comfort/discomfort levels of the ET index (Table A3), it is concluded that the probability of at least uncomfortably 'cold' conditions (ET < 9) dropped from 83% at the beginning to 75% at the ending of the study period, while the probability for at least 'very cold' conditions (ET < 1) decreased from 50% to 42% accordingly. The same index, when applied in Paris, showed a decrease in the 'cold' conditions' probability from 75% to 66% and in the 'very cold' conditions from 42% to 36% between the first and last decade (Figure 9h). In Sofia, the probability of cold-related discomfort conditions based on WCT (WCT < −10) dropped from 7.5% to 3% between the first and last decades (Figure 9i), while at Oulou, cold discomfort conditions based on WCT decreased from 38% to 28%. The decrease is more prominent in the probability of cold-related risk (WCT < −28), which declined from 1.7% in 1976–1985 to 0.4% in 2006–2015 at Oulou.

## 4. Discussion and Conclusions

The analysis highlighted some very important features of European cities with different local climates with regards to the multiannual variability and trends in the mean climate, the frequency

of extreme temperatures, and changes in the heat- and cold-related thermal discomfort. Without exception, all cities have undergone significant warming in all seasons, during the common period 1976–2018. The rate of warming is differentiated between cities and seasons, as well as between $T_{max}$ and $T_{min}$. In almost all cases, the increasing trends in $T_{max}$ are higher during spring and summer, exceeding 0.4 °C/decade. In winter, warming rates increase with increasing latitude. Cities with colder local climates like Helsinki and Oulou become markedly less cold in winter and autumn, with the warming rates in $T_{max}$ exceeding 0.7 °C/decade since the mid-1970s. The warming trends are even more striking in the winter $T_{min}$ though, exceeding 1 °C/decade and reaching up to 1.4 °C/decade since the mid-1970s. These results are consistent between all stations in Finland (Helsinki-K, Helsinki-V, and Oulou). The opposite pattern is observed in warmer cities of lower latitudes like Athens and Palma, with winter $T_{max}$ trends being much lower (and not statistically significant) compared to summer and spring ones, which are very prominent especially in $T_{min}$, amounting to nearly 0.8 °C/decade. Sofia exhibits outstanding warming rates in $T_{max}$ during all seasons and reaches 1.2 °C/decade in summer. Interestingly, a marginally negative trend in winter $T_{min}$ is observed in the city. Central stations like Paris and Rotterdam present lower and not statistically significant trends in winter and autumn $T_{min}$ as well.

The analysis also revealed an increase in the frequency of hot extremes (both hot days and HWs) and decrease in the cold extremes (cold days and CWs) frequency at all cities. Hot extremes in most cities have almost tripled during the last two or three decades compared to the past ones. It is noteworthy that northern cities in Finland have experienced an unprecedented increase in the HWs frequency only during the last decade in contrast to the other cities presenting an earlier beginning of increased hot extremes around the 1990s (Paris, Rotterdam, Sofia, Athens). However, the decline in CWs frequency is gradual during the last four decades at the same cities. Sofia exhibits an overwhelming increase in the hot days and HWs frequency during the last two decades with a nearly four-fold increase in their frequency compared to the past decades. Despite the general pattern of increasing (decreasing) trend in hot (cold) extremes in all cities, not all of the trends are statistically significant (Table A7), given the fact that HWs/CWs are, in general, rare events and the time series do not span many decades in some stations (Table A7).

In addition to the above trends, the analysis unveiled some robust findings with regards to the timing of hot extremes, with earlier occurrence by several weeks of the first hot extreme in the year but also extension (less prominent) of the last hot extreme. Earlier occurrence was statistically significant ($p < 0.05$) at all cities except Rotterdam and Oulou, and later occurrence was statistically significant at southeastern cities (Athens and Sofia), also resulting in a prominent expansion of the hot extremes' season at the beginning and ending of summer.

The use of bioclimatic indices valid for heat and/or cold stress suggested a robust tendency towards less cold-related stress (mainly at cold cities) and more heat-related stress at all cities. The relative decline in the frequency of cold-related discomfort during the last decade compared to the 1970s ranged between 20% and 60% in colder cities (Helsinki-V, Oulou, Paris, Rotterdam, Sofia), while the corresponding frequency in high cold-related risk at northern cities decreased by 36% at Oulou and 45% at Helsinki-V.

Heat stress conditions indicated by HI and HD present a significant upward trend between 1976 and 2018 at all cities as well, which is stronger at southern cities (Athens, Palma, Sofia), with four-fold or even five-fold increases in the frequency of at least 'extreme caution' conditions. In the cities of higher latitudes like Oulou and Helsinki, 'caution' conditions (based on HI) and 'discomfort' conditions (based on HD) are also becoming more frequent and have almost quadrupled during the last decade. Such conditions are almost missing in the 1970s and 1980s.

The study focused on European cities with very different background climate, city size, population, or urbanization levels. It is expected that the observed trends are the combined result of both global/regional warming and urban effect. The contribution of these factors on the observed trends might also be different between the cities, given the different rates of population growth or urban

expansion over the study periods. For instance, increasing UHI intensity in Athens has been reported to account for approximately half of the observed trends in the annual temperature over the period 1970–2004 [55]. However, it is beyond the scope of this study to investigate the causes behind the observed trends.

Overall, the analysis highlighted the ongoing increasing exposure of European population at any geographical area and any latitude to serious heat-related risk forced by climatic factors such as profound increasing trends in the mean air temperature and frequency of temperature hot extremes, but also advancing of hot weather and expansion of the hot extremes' season.

Other socioeconomic factors such as population aging and ongoing urbanization may further increase European population vulnerability to thermal risk in the future. Although higher frequencies and levels of heat stress are observed in cities with warmer climates, the increasing frequency of heat-related discomfort conditions in the central and northern Europe could potentially be more influential to human health due to lack of acclimatization and subjectivity of thermal perception in populations living in colder climates. Yet, adopting different metrics or hazardous indices, recent research points to a further increase in heat-related risk in the coming decades for European cities. While capital cities of the Eastern Mediterranean have been identified as hot spots with respect to future heat-related risk [57–60], other cities of central Europe may even experience a doubling in thermal discomfort hours by 2050 as well [61].

The study points to the imperative need of taking measures to manage thermal risk in Europe and prevent the devastating effects of such fundamental threat on human health.

**Author Contributions:** Conceptualization, D.F. and I.K.; methodology, D.F. and G.K.; software and analysis, F.P. and G.K.; writing—review and editing, D.F.; funding acquisition, I.K.

**Funding:** This research was funded by European Commission's Directorate-General for European Civil Protection and Humanitarian Aid Operations, grant number 783180.

**Conflicts of Interest:** The authors declare no conflict of interest.

## Appendix A

### *Appendix A.1. Heat Index (HI)*

The HI defines an apparent temperature of thermal sensation, combining air temperature and relative humidity and is broadly used by the National Weather Service of the United States for heat-related warning alarms [45]. The HI calculations are based on multiple regression analysis using the following formula [62,63], where T is the air temperature (in °C) and RH the relative humidity (in %). The formula is meaningful when T > 20 °C.

$$
\begin{aligned}
HI = &-8.784695 + 1.61139411 \times T + 2.338549 \times RH - 0.14611605 \times T \times RH - \\
&1.2308094 \times 10^{-2} \times T^2 - 1.6424828 \times 10^{-2} \times RH^2 + 2.211732 \times 10^{-3} \times T^2 \times RH + \\
&7.2546 \times 10^{-4} \times T \times RH^2 - 3.582 \times 10^{-6} \times T^2 \times RH^2,
\end{aligned} \tag{A1}
$$

The assessment scales of HI in terms of heat disorders and degree of comfort, respectively, are presented in the following table.

**Table A1.** Classification of heat index (HI) values and relevant effects on populations.

| Heat Index (°C) | Classification | General Effect on People in High Risk Groups |
|---|---|---|
| ≥54 | Extremely hot—Extreme danger | Heat/Sunstroke highly likely with continued exposure |
| 41–54 | Very hot—Danger | Sunstroke, heat cramps and exhaustion likely, possible heatstroke with prolonged exposure and/or activity |
| 32–41 | Hot—Extreme caution | Sunstroke, heat cramps, or heat exhaustion possible with prolonged exposure and/or physical activity |
| 27–32 | Very warm—Caution | Fatigue possible with prolonged exposure and/or physical activity |

*Appendix A.2. Humidex (HD)*

The HD describes the physical distress of an average person under intense heat and high vapor pressure conditions [64]. The HD was first introduced and is broadly used by environmental and weather services in Canada. The formula for the calculation of HD is presented below, combining air temperature and vapor pressure. The formula is valid when air temperature is higher than 21 °C, where T is the air temperature (in °C) and $e$ is the vapor pressure (in hPa):

$$HD = T + \frac{5}{9} \times (e - 10), \tag{A2}$$

The assessment scales HD in terms of degree of comfort are presented in Table A2.

**Table A2.** Levels of human discomfort according to humidex (HD) values.

| Humidex (°C) | Degree of Comfort |
|:---:|:---:|
| >45 | Dangerous; heat stroke possible |
| 40–45 | Great discomfort; avoid exertion |
| 30–39 | Some discomfort |
| 20–29 | Little discomfort |

*Appendix A.3. Effective Temperature (ET)*

In addition to the air temperature and relative humidity, ET also involves the wind speed (v). The following formulas were used for the calculation of ET (T in °C, RH in %, v in m/s at 1.2 m above the ground).

$$ET = 37 - \frac{(37 - T)}{\left(0.68 - 0.0014 \times RH + \frac{1}{1.76 + 1.4 \times v^{0.75}}\right)} - 0.29 \times T \times (1 - 0.01 \times RH), \tag{A3}$$

Valid for v > 0.2 m/s

$$ET = T - 0.4 \times (T - 10) \times (1 - 0.01 \times RH), \tag{A4}$$

Valid for v < 0.2 m/s

The scales of thermal sensation according to ET [65] are presented in Table A3.

**Table A3.** Scales of thermal sensation according to effective temperature (ET).

| Thermal Sensation | ET Scale (°C) |
|:---:|:---:|
| Very cold | <1 |
| Cold | 1–9 |
| Cool | 9–17 |
| Fresh | 17–21 |
| Comfortable | 21–23 |
| Warm | 23–27 |
| Hot | >27 |

*Appendix A.4. New Wind Chill Equivalent Temperature (WCT)*

The index is valid only for cold stress and is calculated using the following formula

$$WCT = 13.12 + 0.6215 \times T - 11.37 \times v^{0.16} + 0.3965 \times T \times v^{0.16}, \tag{A5}$$

Air temperature (T) is in °C and wind speed (v) in km/h, at 10 m above ground level. The index is valid only for T ≤ 10 °C and v ≥ 5 km/h [66]. Table A4 presents the scales of thermal sensation and related health hazards for WCT [65,67].

**Table A4.** Scales of thermal sensation and related hazards according to new wind chill equivalent temperature (WCT).

| Related Health Hazards | Interval (°C) | Thermal Sensation |
|---|---|---|
| No discomfort | >0 | Comfortable |
| Slight discomfort | 0 to −9 | |
| Uncomfortable | −10 to −27 | Cool |
| Risk | −28 to −39 | Cold |
| High risk | −40 to −47 | Very cold |
| Very high risk | −48 to −54 | |
| Dangerous | ≤−55 | Frosty |

## Appendix B

**Table A5.** Statistical significance (*p*-values) of the seasonal trends in $T_{max}$ using the Man–Kendall test. Italic entries denote not statistically significant trends based on 95% confidence level.

| Stations | Winter | Spring | Summer | Autumn |
|---|---|---|---|---|
| Athens (NOA) | *1.24 × 10⁻¹* | $1.52 \times 10^{-3}$ | $1.28 \times 10^{-5}$ | $3.54 \times 10^{-2}$ |
| Palma | *1.09 × 10⁻¹* | $8.69 \times 10^{-5}$ | $9.79 \times 10^{-4}$ | $2.72 \times 10^{-2}$ |
| Sofia | $6.52 \times 10^{-3}$ | $4.11 \times 10^{-6}$ | $6.38 \times 10^{-8}$ | $2.05 \times 10^{-5}$ |
| Paris | $4.98 \times 10^{-2}$ | $6.12 \times 10^{-5}$ | $7.81 \times 10^{-4}$ | $3.74 \times 10^{-3}$ |
| Rotterdam | $4.13 \times 10^{-2}$ | $1.13 \times 10^{-4}$ | $9.79 \times 10^{-4}$ | $6.21 \times 10^{-4}$ |
| Helsinki-K | $2.87 \times 10^{-2}$ | $9.48 \times 10^{-5}$ | $1.52 \times 10^{-3}$ | $2.20 \times 10^{-4}$ |
| Helsinki-V | $3.54 \times 10^{-2}$ | $1.03 \times 10^{-4}$ | $2.17 \times 10^{-3}$ | $2.03 \times 10^{-4}$ |
| Oulou | $1.03 \times 10^{-2}$ | $1.75 \times 10^{-3}$ | $9.16 \times 10^{-3}$ | $5.12 \times 10^{-5}$ |

**Table A6.** As in Table A5 but for $T_{min}$.

| Stations | Winter | Spring | Summer | Autumn |
|---|---|---|---|---|
| Athens (NOA) | $3.03 \times 10^{-2}$ | $1.40 \times 10^{-5}$ | $3.81 \times 10^{-9}$ | $1.41 \times 10^{-5}$ |
| Palma | $3.19 \times 10^{-2}$ | $3.48 \times 10^{-11}$ | $1.56 \times 10^{-9}$ | $1.40 \times 10^{-6}$ |
| Sofia | *7.37 × 10⁻¹* | $8.45 \times 10^{-3}$ | $5.05 \times 10^{-6}$ | *6.08 × 10⁻²* |
| Paris | *1.56 × 10⁻¹* | $5.55 \times 10^{-3}$ | $1.31 \times 10^{-3}$ | $7.61 \times 10^{-3}$ |
| Rotterdam | *1.24 × 10⁻¹* | $1.63 \times 10^{-3}$ | $9.56 \times 10^{-6}$ | *1.46 × 10⁻¹* |
| Helsinki-K | $3.27 \times 10^{-3}$ | $2.19 \times 10^{-2}$ | $4.28 \times 10^{-3}$ | $1.58 \times 10^{-4}$ |
| Helsinki-V | $3.27 \times 10^{-3}$ | $5.92 \times 10^{-3}$ | $3.46 \times 10^{-7}$ | $1.40 \times 10^{-5}$ |
| Oulou | $9.08 \times 10^{-4}$ | *2.72 × 10⁻¹* | $4.56 \times 10^{-2}$ | $3.88 \times 10^{-4}$ |

**Table A7.** Slopes in the average number of hot/cold and HWs/CWs per decade at the selected stations, along with *p*-values. Asterisks (**) denote statistical significance at 95% confidence level and (*) denote statistical significance at 90% confidence level.

| Station | Period | Hot Days/Decade (*p*-Value) | HWs/Decade (*p*-Value) | Cold Days/Decade (*p*-Value) | CWs/Decade (*p*-Value) |
|---|---|---|---|---|---|
| Athens (NOA) | 1897–2016 | +0.60 ** (0.0000) | +0.09 ** (0.0000) | −0.42 ** (0.0007) | −0.06 ** (0.0042) |
| Palma | 1979–2018 | +0.47 (0.4621) | +0.10 (0.3955) | −1.56 ** (0.0006) | −0.16 ** (0.0344) |
| Sofia | 1979–2018 | +4.94 ** (0.0004) | +0.73 ** (0.0008) | - | - |
| Paris | 1949–2018 | +0.78 ** (0.0043) | +0.06 (0.1551) | −1.64 ** (0.0055) | −0.26 ** (0.0052) |
| Rotterdam | 1959–2018 | +0.87 ** (0.0020) | +0.12 ** (0.0144) | −0.84 * (0.0563) | −0.15 ** (0.0448) |
| Helsinki-K | 1959–2018 | +0.62 (0.1545) | +0.08 (0.2369) | −0.67 * (0.0827) | −0.08 (0.1887) |
| Helsinki-V | 1959–2018 | +1.08 ** (0.0105) | +0.15 ** (0.0391) | −0.74 ** (0.0359) | −0.09 * (0.0929) |
| Oulou | 1959–2018 | +0.41 (0.3473) | +0.12 * (0.0834) | −0.72 * (0.0609) | −0.07 (0.3615) |

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
