# Peer review of "Observed Trends in Thermal Stress at European Cities with Different Background Climates"

_atmosphere, doi:10.3390/atmos10080436_

Round 1
Reviewer 1 Report
The study analyses trends and changes in mean air temperature, frequency of extreme temperatures, and bioclimatic indices (measuring heat and cold‐related thermal discomfort) in 7 European cities based on historical observations, mainly for the period 1976‐2018. Even though the study does not cover many locations and a long analysis period, the results contribute to the current knowledge on urban thermal risk. The paper is very well written, and the contents are clear and understandable by a large audience. My recommendation is Accept after minor revision, and the following are recommendations for improvements.
1) Section 2.3 should explain why authors selected two different weather stations in Helsinki. The discussion of results should mention the fact that Helsinki-V is located further away from the city centre and at a higher altitude than Helsinki-K. Hence, Helsinki-V might not capture well the exposure of urban population to thermal risk. Whenever there is available data for Helsinki-K and Helsinki-V in the period 1979‐2018, the results should be presented for both locations and compared.
2) Figure 1 is not informative due to the high inter-annual variability, and to the number of time-series included. I highly recommend replacing it with a graphic of a moving-average of the data (e.g., 5-year or 10-year moving average), which better highlights the long-term trends.
See for example Figure 6 of Costa & Soares (2009), or Figs. 6-10 of Kousari, Ahani, & Hendi-zadeh (2013):
Costa, A. C., & Soares, A. (2009). Trends in extreme precipitation indices derived from a daily rainfall database for the South of Portugal. International Journal of Climatology, 29(13), 1956-1975. https://doi.org/10.1002/joc.1834
Kousari, M. R., Ahani, H., & Hendi-zadeh, R. (2013). Temporal and spatial trend detection of maximum air temperature in Iran during 1960–2005. Global and Planetary Change, 111, 97-110. https://doi.org/10.1016/j.gloplacha.2013.08.011
3) In Figure 2, the order of the cities in the horizontal axis should be the same in graphs a) and b).
4) Why is Sofia missing from graph b) in Figure 2?
In Figure 2, as well as in other analysis, I suggest reducing the analysis period to 1979‐2018 whenever this allows to show the results of more locations.
5) I suggest adding a Table to Appendix B showing the station’s name, the analysed period, the slope, and the p-value of the trends in the Frequency of hot /cold days per decade and HWs /CWs per decade. This table should cover all stations with data at least in the 1979‐2018 period. Reference to this table should be made in section 3.2.
6) Review text in lines 357-361. According to Table A1, ‘Caution’ corresponds to HI values between 27 and 32 (not all cases above 27), and ‘Extreme caution’ to HI values between 32 and 41 (not all cases above 32). Similarly, for the description of the HD index classes.
7) In Figure 7 it is not clear if ‘disconfort’ corresponds to values classified as ‘Little discomfort’ or ‘Some discomfort’, or both. Please clarify in the legend.
8) The legend of all Figures and Tables (including those in Appendix A) referring to bioclimatic indices must indicate the name of the index and the corresponding abbreviation. For example, in Figure 7. (…) Humidex (HD) classification (…).
9) Section 3.4.3 should discuss all the results related to the CDFs of bioclimatic indices; not only those illustrated in Figure 8.
In fact, little or nothing is said about non-significant trends or lack of changes throughout the discussion of results. Such results should also be shown, or at least they should be highlighted in the discussion.
10) Section 4. Discussion & Conclusions would benefit from a comparison between the results of Helsinki-K and those of Helsinki-V. Are they identical? If not, why not?
Moreover, I recommend pointing out that this study does not assess the causes of the observed trends, which may be related to global climate change and/or to an increased UHI effect.
Finally, the following are minor improvements to text and References:
Table 2: Indicate the analysis period in the legend.
Table 2: Helsinki-K or Helsinki-V?
Table 3: Indicate the analysis period in the legend.
Table 3: Helsinki-K or Helsinki-V?
Lines 13-14 (Abstract): decades, demonstrated the increased vulnerability of population even in developed countries from Europe, which are expected to manage efficiently adverse weather.
Lines 35-36: Make a proper citation to the IPCC Report as indicated in p. 49 of the PDF. A link can be provided in the text, but I recommend the following one: https://www.ipcc.ch/sr15/
Line 40: possibly causing irreversible changes ‐ and also to humans
Line 41: Changes in the mean climate are accompanied by changes in climatic extremes
Line 76: rate in cities even by four times when compared to rural areas
Line 96: The present study focuses on seven European cities
Line 119: The differences in the percentiles
Line 121: It is thus relevant to study climate variability
Line 206: … for Sofia, the corresponding results are
Line 209: exhibit
Line 224: The year 2018 was the warmest year of the historical record of Tmin in Athens.
Line 225: Palma and Rotterdam experienced higher warming rates
Line 239: All cities experience large increases
Line 313: upper percentile of the valid values (???) for heat stress
Line 315: lower percentile of the valid values (???) for cold stress indices
Line 323: Regarding the cold‐related stress
Lines 323-324: percentile of the valid values (???) for cold indices (ET and WCT)
Line 357: are depicted in Figs. 6 and 7 [the point is missing after Figs]
Line 366: (Fig. 6) [the point is missing]
Line 404: Figure 8 (c, d)
Line 405: depicts CDF curves
Line 434: outstanding warming rates in Tmax during all seasons, and reaches 1.2 °C/decade in summer.
Line 435: Τhe analysis also revealed a profound increase
Line 444: robust findings as regards to the
Lines 454-455: present a significant upward trend between 1976 and 2018 at all cities
Line 485 (Appendix A): National Weather Service of the United States for
Line 500 (Appendix A): and is broadly used by Environmental
Author Response
Answers to reviewer 1
The authors are grateful to reviewer 1 for his/her time devoted to this paper and the helpful suggestions and comments aiming at the improvement of the study. We tried to revise our manuscript accounting for the suggested recommendations to the best possible degree.
‘The study analyses trends and changes in mean air temperature, frequency of extreme temperatures, and bioclimatic indices (measuring heat and cold‐related thermal discomfort) in 7 European cities based on historical observations, mainly for the period 1976‐2018. Even though the study does not cover many locations and a long analysis period, the results contribute to the current knowledge on urban thermal risk. The paper is very well written, and the contents are clear and understandable by a large audience. My recommendation is Accept after minor revision, and the following are recommendations for improvements.’
We are very happy to know that reviewer finds this research interesting. In the following we provide answers to each one of the reviewer’s comments.
‘1) Section 2.3 should explain why authors selected two different weather stations in Helsinki. The discussion of results should mention the fact that Helsinki-V is located further away from the city centre and at a higher altitude than Helsinki-K. Hence, Helsinki-V might not capture well the exposure of urban population to thermal risk. Whenever there is available data for Helsinki-K and Helsinki-V in the period 1979‐2018, the results should be presented for both locations and compared.’
We thank the reviewer for this comment. Indeed, we used two different stations in Helsinki ( Helsinki-Kaisianemi and Hesinki-Vantaa). Actually, we extracted daily air temperature data for these stations from the Finnish Meteorological Institute (as described in Methods) and we found it useful to benefit from their long and complete records. Helsinki-K is an urban station and Helsinki-V is the airport station, a few kilometres to the north and at low altitude also. However, since Helsinki-K is located at a green place almost next to the sea, summer temperatures at this station are lower compared to Helsinki-V (see also relevant percentiles in Table 1 of the manuscript). In this respect, Helsinki-K may not act as a station with strong UHI effect. Moreover, Helsinki-V is also affected by urbanization and built surfaces (visual inspection from Google maps). It also notable that seasonal trends at Helsinki-V are higher compared to Helsinki-K which is also highlighted in the text. Both stations are presented also in the (added) Tables B1-B3.
Regarding the analysis of bioclimatic indices, we used only Helsinki-V because this was the only station with available synchronous measurements at 3-hours resolution, necessary for the analysis of thermal stress.
‘2) Figure 1 is not informative due to the high inter-annual variability, and to the number of time-series included. I highly recommend replacing it with a graphic of a moving-average of the data (e.g., 5-year or 10-year moving average), which better highlights the long-term trends.
See for example Figure 6 of Costa & Soares (2009), or Figs. 6-10 of Kousari, Ahani, & Hendi-zadeh (2013):
Costa, A. C., & Soares, A. (2009). Trends in extreme precipitation indices derived from a daily rainfall database for the South of Portugal. International Journal of Climatology, 29(13), 1956-1975. https://doi.org/10.1002/joc.1834
Kousari, M. R., Ahani, H., & Hendi-zadeh, R. (2013). Temporal and spatial trend detection of maximum air temperature in Iran during 1960–2005. Global and Planetary Change, 111, 97-110. https://doi.org/10.1016/j.gloplacha.2013.08.011’
We thank the reviewer for this comment and we fully agree with it. We replaced the figure with a new one, using a 5-year moving average filter which improves readability of the figure and reduces to some extent the high inter-annual variability.
‘3) In Figure 2, the order of the cities in the horizontal axis should be the same in graphs a) and b).’
We thank the reviewer for this comment. Actually the order is the same in a) and b) graphs but it looks different because Sofia station is missing from the analysis related to Tmin. This is because there were large gaps in the time series of the daily Tmin for this station. This was already mentioned in the manuscript (Line 206-207 in pdf of the initial submission). However, we finally decided to add seasonal Tmin trends for Sofia, calculating monthly values of Tmin from the available monthly values of Tavg and Tmax and the commonly used formula Tavg= (Tmax+Tmin)/2, also recommended by World Meteorological Organization (WMO). (e.g. WMO, 2008.(added in Reference list) . We have now replaced the relevant Figures keeping the same order of the stations (as in Table 1). We also changed the plot to a bar plot allowing the inclusion of confidence intervals.
4) Why is Sofia missing from graph b) in Figure 2?
Please see the answer in the previous comment
‘In Figure 2, as well as in other analysis, I suggest reducing the analysis period to 1979‐2018 whenever this allows to show the results of more locations.’
We thank the reviewer for this comment. The available data at the selected stations in Table 1 cover periods at least since 1974. We decided to use the period from the mid1970s onwards (actually 1976) in seasonal analysis as this is widely aknowledged ‘climate shift’ year (lines 190-191 in pdf) but also allows to include all selected stations in Table1. In this respect, the use of 1979-2018 period, wouldn’t allow to show more stations. The period 1979-2018 was used when the analysis was performed on decades resolution to ensure four decades.
‘5) I suggest adding a Table to Appendix B showing the station’s name, the analysed period, the slope, and the p-value of the trends in the Frequency of hot /cold days per decade and HWs /CWs per decade. This table should cover all stations with data at least in the 1979‐2018 period. Reference to this table should be made in section 3.2.’
We thank the reviewer for this comment. We have added the table (Table B3) in the Appendix B, showing the suggested information for the whole available period for each station and for at least the 1979-2018 period.
‘6) Review text in lines 357-361. According to Table A1, ‘Caution’ corresponds to HI values between 27 and 32 (not all cases above 27), and ‘Extreme caution’ to HI values between 32 and 41 (not all cases above 32). Similarly, for the description of the HD index classes.’
We thank the reviewer for this comment and we fully agree for this recommendation. Actually all values > 27 (or > 32 etc) were used in the analysis, so the word ‘caution’ is indeed misleading based on the comfort scales according to Tables in Appendix A. The ‘caution’ indication is now replaced with ‘at least caution;, ‘at least discomfort’ etc in the text, figures, legends. We apologize for not clarifying it earlier.
7) In Figure 7 it is not clear if ‘disconfort’ corresponds to values classified as ‘Little discomfort’ or ‘Some discomfort’, or both. Please clarify in the legend.
We thank the reviewer for this comment. The values always refer to HD> 29, (which is already mentioned in the text and is also added in the legend of the figure now) which strictly refers to at least ‘some discomfort’, however, we preferred the general ‘at least discomfort’ term to highlight the lack of comfortable conditions.
8) The legend of all Figures and Tables (including those in Appendix A) referring to bioclimatic indices must indicate the name of the index and the corresponding abbreviation. For example, in Figure 7. (…) Humidex (HD) classification (…).
We thank the reviewer for this comment. We have applied necessary corrections.
9) Section 3.4.3 should discuss all the results related to the CDFs of bioclimatic indices; not only those illustrated in Figure 8.
We thank the reviewer for this comment. Figure 8 (which corresponds to Fig. 9 in the new manuscript), has been revised. We have added also some results based on the Effective Temperature (ET) CDFs and in general more information regarding the decrease in probabilities of cold related stress. The Figure now includes indicative results related to changes in heat-related and cold-related stress. The discussion is enriched with additional results related to the CDFs of bioclimatic indices.
‘In fact, little or nothing is said about non-significant trends or lack of changes throughout the discussion of results. Such results should also be shown, or at least they should be highlighted in the discussion.’
We thank the reviewer for this comment. We tried to add more information about results in all stations and even not statistically significant changes in several parts of the text, and also in Discussion. Tables B1-B3 have been also added in a new Appendix (B) showing significance levels of all stations for seasonal trends and trends in hot/cold extremes.
10) Section 4. Discussion & Conclusions would benefit from a comparison between the results of Helsinki-K and those of Helsinki-V. Are they identical? If not, why not?
We thank the reviewer for the comment. As already mentioned, only Helsinki-V was used for the assessment of heat stress based on the bioclimatic indices (related to the availability of synchronous 3-hourly data) , so no comparison is possible for these indices. Regarding the climatic indices, results between the two stations are very consistent as it can be concluded from the seasonal trends (Fig. 3), but also trends in hot/cold extremes (e.g Table B3). The results are consistent but not identical. As a matter of fact it is not possible to find identical results even between neighbour stations, since even factors on very local or microscale might be influential. The stations are at distances of some kilometres but surrounding environments are different as for instance the proximity of Helsinki-K to the sea which makes the station cooler in summer compared to Helsinki-V. We did not make an extensive discussion with respect to the comparison between these two stations, since the study mainly focused on the assessment of trends at urban areas and not the causes behind this (see also the next comment).
‘Moreover, I recommend pointing out that this study does not assess the causes of the observed trends, which may be related to global climate change and/or to an increased UHI effect.’
We thank the reviewer for this comment. We have added a relevant sentence at the ending of ‘Introduction’.
Finally, the following are minor improvements to text and References:
Table 2: Indicate the analysis period in the legend.
The period is added in the legend
Table 2: Helsinki-K or Helsinki-V?
This is clarified in the Table
Table 3: Indicate the analysis period in the legend.
The period is added in the legend
Table 3: Helsinki-K or Helsinki-V?
This is clarified in the Table
Lines 13-14 (Abstract): decades, demonstrated the increased vulnerability of population even in developed countries from Europe, which are expected to manage efficiently adverse weather.
This is corrected
Lines 35-36: Make a proper citation to the IPCC Report as indicated in p. 49 of the PDF. A link can be provided in the text, but I recommend the following one: https://www.ipcc.ch/sr15/
We thank the reviewer for this comment. We have replaced the link in the text and we have added a proper citation in the reference list.
Line 40: possibly causing irreversible changes ‐ and also to humans
This is corrected
Line 41: Changes in the mean climate are accompanied by changes in climatic extremes
This is corrected
Line 76: rate in cities even by four times when compared to rural areas
This is corrected
Line 96: The present study focuses on seven European cities
This is corrected
Line 119: The differences in the percentiles
This is corrected
Line 121: It is thus relevant to study climate variability
This is corrected
Line 206: … for Sofia, the corresponding results are
This is corrected
Line 209: exhibit
This is corrected
Line 224: The year 2018 was the warmest year of the historical record of Tmin in Athens.
This is corrected
Line 225: Palma and Rotterdam experienced higher warming rates
This is corrected
Line 239: All cities experience large increases
This is corrected
Line 313: upper percentile of the valid values (???) for heat stress
This is corrected
Line 315: lower percentile of the valid values (???) for cold stress indices
This is corrected
Line 323: Regarding the cold‐related stress
This is corrected
Lines 323-324: percentile of the valid values (???) for cold indices (ET and WCT)
This is corrected
Line 357: are depicted in Figs. 6 and 7 [the point is missing after Figs]
This is corrected
Line 366: (Fig. 6) [the point is missing]
This is corrected
Line 404: Figure 8 (c, d)
This is corrected
Line 405: depicts CDF curves
This is corrected
Line 434: outstanding warming rates in Tmax during all seasons, and reaches 1.2 °C/decade in summer.
This is corrected
Line 435: Τhe analysis also revealed a profound increase
This is corrected
Line 444: robust findings as regards to the
This is corrected
Lines 454-455: present a significant upward trend between 1976 and 2018 at all cities
This is corrected
Line 485 (Appendix A): National Weather Service of the United States for
This is corrected
Line 500 (Appendix A): and is broadly used by Environmental
This is corrected
(In addition to changes in the text, we have applied changes in Figures, while Tables B1-B3 were added in Appendix B)
Figure 1: (a map was added)
Figure 2 (previous Fig. 1): A 5-year moving average was applied
Figure 3: The plot changed to bar plot and confidence intervals were added
Figure 5-8: minor corrections/editing
Figure 9: More CDF plots were added
Reviewer 2 Report
Please see the attachment.

Author Response
Answers to Reviewer 2
The authors are grateful to the reviewer for his/her time devoted to this research study and the very useful comments and suggestions aiming at the improvement of the paper. We tried to revise the manuscript accounting for all recommendations. In the following, we provide answers to each one of the reviewer’s comment.
‘The article deals with the increasing temperatures and occurrence of heat waves and cold waves in several cities in Europe. Even though temperatures have been thoroughly explored in previous studies, this paper also focuses on thermal stress evaluation by evaluating several bioclimatic indices.
The manuscript is well structured, with coherent sections. Well-presented results that will be highly cited in future studies, particularly section 3.4.2. However, due to the different time periods investigated for each station I am concerned about the comparison between the stations. ‘
We are very happy to know that the reviewer finds this research interesting. Indeed, available climatic data cover different periods in each station, whereas long enough to highlight trends analysis spanning some decades. Although we found interesting to show the variation of certain variables over the whole available period for each stations (e.g. the centennial variations at Athens), when comparisons were attempted (e.g. analysis of seasonal trends), a common period was assumed (please see also the answer in a next comment). In most parts of the analysis, a common period is used between the stations (e.g. Figs 3, 5,6,7,8,9) or Tables 2, 3.
‘Line 107. Provide a map with the location of the eight stations.’
We thank the reviewer for this comment. We have now added a relevant map indicating the stations sites (Fig. 1 in the revised manuscript).
‘Line 114. I understand that the data you optimized for the eight stations depend on the availability of data for each station. In section 2.1 you found the percentiles (5th and 95th) based on the available datasets of each region. How can the maximum and minimum temperatures thresholds be comparable if you used different time periods for each station? Climatic change suggests that the mean temperature increased from the previous century (19th century) or the beginning of the 20th century. So the 95th percentile in Athens cannot be compared with the other stations. ‘
We thank the reviewer for this comment. We fully agree that the percentiles cannot be comparable if different periods are considered. However, we did not use different periods for this estimation, but a common period (1974-2003) for all stations as already indicated in the text (Line 114) of our initial submission and this also added in Table 1 caption in the revised version. Although the period 1970-2000 is commonly used as reference period in climatic studies, the selected period in our study was imposed by data availability and reflects the earlier common 30-yrs period in our data sets.
‘Line 151. Why did you choose these specific bioclimatic indices? Were they used in previous studies? Is it because of the availability of only temperatures, RH and wind? Why did you study them for every 3-hours? ‘
We thank the reviewer for these comments. Actually, there is a plethora of bioclimatic indices in literature, simple or rational thermal indices. The selection of the indices was mainly imposed by the availability of the involved meteorological variables, while they are broadly used by weather services worldwide. Other indices, as for instance the more recent Universal Thermal Climate Index (UTCI) requires solar radiation data and thus could not be used in this study. Despite limitations, or simplicity of the indices, our analysis mainly focuses on the long-term ‘trends’ and of the selected indices and comparison between cities with different background climate.
As regards the use of 3-hours resolution: The estimation of the bioclimatic indices requires ‘simultaneous’ measurements of the involved meteorological measurements to assess the thermal comfort level at a particular time. The use of daily measurements is not appropriate as it provides ‘average’ values of T, RH, wind speed etc which by no means represent actual thermal conditions. As regards hourly values, these were available only in certain stations, while in other stations only for limited periods. We decided to use the 3-hours interval as this was the shortest ‘common’ time interval for all stations with ‘simultaneous’ measurements.
‘Section 3.1. Figure 1:
How did you calculate the temperature anomaly? Is it monthly anomaly or yearly? Please provide explanation of the calculation in the methodology. ‘
We thank the reviewer for this comment. We fully agree that additional information should be provided as regards the calculation of the air temperature anomaly. The plots in the figure show annual temperature anomalies calculated as differences from the 1974-2003 mean, (which is also the reference period already used in the calculation of percentiles in Table 1). This is now clarified in the text but also in the Figure caption.
‘Figure 2:
How was the seasonal trend evaluated? Provide explanation in methodology. Why don’t you use Mann-Kendall seasonal analysis? ‘
We thank the reviewer for this comment. These are linear trends, calculated from a linear regression analysis, as already mentioned in methodology of the initial submission (Line 127). We have now used the Mann-Kendall test to investigate the statistical significance in the seasonal trends (p- values). The results are included in Appendix B (Tables B1 and B2).
‘Also the trends appear to be high. If the maximum temperature increases about 0.1oC/year then in a decade the temperature increases by 1oC? And in a century it will increase by 10oC? Maybe the trend is for decade? ‘
We thank the reviewer for this comment. The estimated trends are really large and do not refer to decades but per year. These large trends are already emphasized in the manuscript, especially as regards the northern cities (Finland). Note that all three stations in Finland for example reveal comparable large trends reflecting pronounced warming rates in winter. Moreover, our estimated trends for Heslinki as regards annual temperatures, are consistent with other recent studies reporting trends of the order of 0.5 0C per decade in the annual temperature over 1980-2018 (e.g. Figures 3, 4 in the study by Agu Eensaar, Climate 2019, 7, 22; doi:10.3390/cli7020022).
In the same study (but also in the centennial record for Athens of our own staudy), century-long trends could be much smaller compared to trends of a certain period, as it is the period after the mid-1970s.
‘Also I would like if the confidence intervals were included in the figure.’
We thank the reviewer for this comment. We changed the chart type to bar plot and we added confidence intervals in the Figure (Fig. 3 in the revised version of the manuscript).
‘Table 3
Put all values to 3d.p.’
This was fixed in the Table
‘Figure 5.
Why did you only choose these four stations to illustrate? Have the same y-axis for the Athens-Sofia pair and for the Rotterdam-Helsinki pair.’
We thank the reviewer for this comment. There is not any very special reason for selecting these stations. We just selected two warm stations to highlight the increase in the upper percentile for heat-related stress and two cold stations to illustrate trends in the lower percentile for cold-related stress. Note that these stations exhibit the largest (and significant) trends, as shown in Table 3. Table 3 presents the trends of all stations, so this information is already included in the paper. However, if the reviewer insists that additional plots from other stations would be more informative, we are willing to add them.
Regarding the y-axis scale, we have homogenized them for Athens and Sofia as well as for Rotterdam and Helsinki pair.
‘There is a peak in HI98, HD98 in Athens for years 1987 and year 2007? Why is that? There is also a drop in ET02, WCT02 for Rotterdam and Helsinki in 1985-1987. Why?’
We thank the reviewer for this comment. Peaks denote some years with very high (low) values of the examined indices. For instance the peaks in 1987 and 2007 for Athens reflect the two severe HWs that Athens experience in 1987 and 2007 (see for instance Founda & Giannakopoulos , Glob. Planet. Change 2009, 67, 227-236. DOI:j.gloplacha.2009.03.013).
‘Section 3.4.3.
Interpret better the meaning of steeper and shift of CDF curves for other cities like you explained it for Athens. ‘
We thank the reviewer for this comment. Figure 8 (which corresponds to Fig. 9 in the new manuscript), has been revised. We have added also some results based on the Effective Temperature (ET) CDFs and in general more information regarding the decrease in probabilities of cold related stress. The Figure now includes indicative results related to changes in heat-related and cold-related stress through. The discussion is enriched with additional results related to the CDFs of bioclimatic indices.
‘Minor comment. Number the figures a-h with the same font and alignment. Also for a and h put the yellow line 1976-1985 over the red one.’
We thank the reviewer for this comment. We have applied the suggested corrections.
(In addition to changes in the text, we have applied changes in Figures, while Tables B1-B3 were added in Appendix B)
Figure 1: (a map was added)
Figure 2 (previous Fig. 1): A 5-year moving average was applied
Figure 3: The plot changed to bar plot and confidence intervals were added
Figure 5-8: minor corrections/editing
Figure 9: More CDF plots were added
Round 2
Reviewer 1 Report
The manuscript has been much improved. My recommendation is to Accept in present form.
Author Response
We are grateful to the reviewer for his/her time devoted to our manuscript and we are very pleased to know that now approves the publication of the revised version
Reviewer 2 Report
the authors replied to all my comments.
Author Response
We would like to thank the reviewer for his/her time devoted to this paper and we are pleased to know he/she is satisfied with the revised version of the manuscript